# Ecological Synthesis of CuO Nanoparticles Using *Punica granatum* L. Peel Extract for the Retention of Methyl Green

**Mongi ben Mosbah** [1,2]**, Abdulmohsen Khalaf Dhahi Alsukaibi** [3]**, Lassaad Mechi** [3]**, Fathi Alimi** [3]
**and Younes Moussaoui** [2,4,]*

[1]  Laboratory for the Application of Materials to the Environment, Water and Energy (LR21ES15),
     Faculty of Sciences of Gafsa, University of Gafsa, Gafsa 2112, Tunisia; mbenmosbah@yahoo.fr
[2]  Faculty of Sciences of Gafsa, University of Gafsa, Gafsa 2112, Tunisia
[3]  Department of Chemistry, College of Science, University of Ha'il, Ha'il 81451, Saudi Arabia;
     a.alsukaibi17@gmail.com (A.K.D.A.); mechilassaad@yahoo.fr (L.M.); alimi.fathi@gmail.com (F.A.)
[4]  Organic Chemistry Laboratory (LR17ES08), Faculty of Sciences of Sfax, University of Sfax, Sfax 3029, Tunisia
*   Correspondence: y.moussaoui2@gmx.fr

**Abstract:** The aqueous extract from the bark of *Punica granatum* L. was invested to generate CuO nanoparticles from $CuSO_4$ using a green, economical, ecological, and clean method. The synthesized nanoparticles were characterized and were successfully used as adsorbents for methyl green retention of an absorptive capacity amounting to 28.7 mg g$^{-1}$. Methyl green equilibrium adsorption data were correlated to the Langmuir model following the pseudo-second order kinetics model. This study clearly corroborates that copper nanoparticles exhibit a high potential for use in wastewater treatment.

**Keywords:** CuO nanoparticles; green synthesis; *Punica granatum* L.; biosorption; methyl green

## 1. Introduction

Currently, nanotechnology is a very interesting field of research due to the successful application of nanomaterials in numerous areas, namely medicine, electronics, environment, biology, and optics [1–4]. Nanoparticles are known for their small size, their large area surface, and their dimensions ranging from 1 to 100 nm [5,6]. The physico-chemical parameters of these materials make them good candidates for a variety of applications such as the removal of pollutants from contaminated waters, biological activities, energy storage systems, etc., [1,7–13]. Numerous techniques have been used to synthesize nanomaterials in the form of particles, colloids, powders, rods, aggregates, tubes, wires, and thin films. These techniques are classified into physical, biological, and chemical methods. Physical methods involve evaporation and mechanical treatments [14,15], whereas chemical methods involve sonochemistry, microemulsion, and electrochemistry [4,14,16]. The latter have the merit of synthesis at a temperature below 350 °C [14,17]. Biological methods rely on the preparation of nanoparticles by such microorganisms as plant extracts, bacteria, yeasts, and fungi [18–20]. These methods have the benefits of easy scaling, non-toxicity, respect for the environment, and reproducibility of manufacturing [19,21]. Thus, the orientation towards natural products gained the widest attention owing to worldwide development, favoring good health, and mitigating the risk of disease. In this respect, the valorization of plant biomass is of paramount importance relating to the protection of the environment [22,23]. Renewable natural resources perform a key role in economic activities [24,25]. In this context, multiple natural resources have been investigated as sources of bioproducts. Components of biomass extracts, including phenols, flavonoids, tannins and other bioproducts, proved to be responsible for reducing metal salts to nanoparticles [13,23,26,27].

CuO nanoparticles are known as a p-type semiconductor [28,29] used as materials for Li batteries, sensors, solar energy, magnetic storage media, semiconductors, catalysis, and adsorption [29–31]. CuO nanoparticles can be synthesized by chemical reduction-based

processes including sol-gel, precipitation, bio-template, microwave heating, electrospinning, and ion exchange [28]. For example, Kumar et al. [32] synthesized CuO nanoparticles from copper nitrate through a hydrothermal reaction and used them for the adsorption of methyl orange from aqueous solution. They found that this synthetic method provided CuO nanoparticles with good adsorption performance. In another study, Darwish et al. [33] prepared CuO nanoparticles through the microwave heating method and the obtained materials were used for the retention of methyl orange. The obtained results indicate that experimental kinetics data correlated with the second-order model and the adsorption was endothermic and spontaneous and governed by a physical process [33]. Moreover, CuO nanoparticles biosynthesized using *Tamarindus indica* extracts exhibit excellent photostability and significant antibacterial activity against bacterial strains [34]. Likewise, Anand et al. [35] showed that CuO nanoparticles synthesized by the microwave combustion method can be used to cure urinary tract infection.

Within this framework, the basic objective of this work lies in using an ecological method for the elaboration of a nanomaterial for the adsorption of aqueous pollutants. In this case, it is a question of examining the possibility of using *Punica granatum* L. bark extract for the preparation of CuO nanoparticles. Indeed, Tunisia is a country with a Mediterranean climate that has several types of plants, among which *Punica granatum* is a drought tolerant species, and it benefits from great capacities to adapt to environmental conditions characterized by marked climatic aridity. The fruits of this plant are rich in secondary metabolites [36–38] which can be used as binding agents to generate nanoscale materials.

## 2. Materials and Methods

### 2.1. Chemicals

Copper sulfate ($CuSO_4 \geq 99\%$), hydrochloric acid (HCl, 37%), sodium hydroxide ($NaOH \geq 98\%$), and methyl green ($C_{23}H_{25}N_3Cl_2.ZnCl_2$) were purchased from Sigma-Aldrich (Tunis; Tunisia). All chemicals were invested as received without any additional purification.

### 2.2. Extract Preparation

The bark extract of *Punica granatum* L. was prepared according to the steps illustrated in Figure 1.

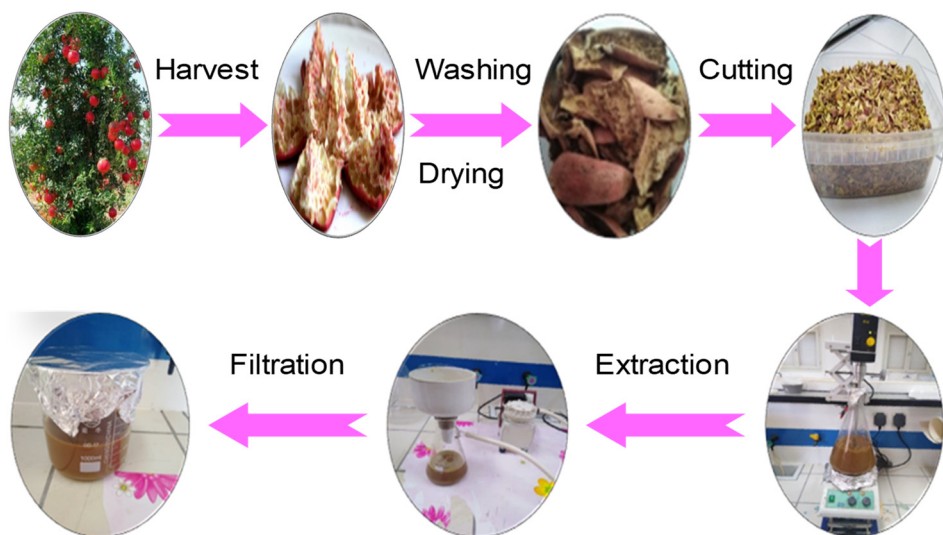

**Figure 1.** Schematic representation of the extract preparation.

*Punica granatum* bark was washed and dried in the shade in a dry and aerated environment at room temperature ($25 \pm 2$ °C). After drying, the material was cut into small pieces. To prepare the extracts, 200 g of obtained material were mixed with 800 mL of distilled

water. The obtained mixture underwent mechanical stirring at a temperature between 50 and 60 °C, until we obtained a pasty solution. Subsequently, the mixture was filtered using Whatman filter paper No.01, before it was recovered and stored at the temperature of 4 °C.

### 2.3. Chemical Composition of Punica granatum Bark

Chemical composition of *Punica granatum* bark was specified according to standardized methods. The ash content was estimated according to the T211 om-07 method. A given mass ($m_0$) of the material was calcined in an oven at 600 °C for 6 h to provide a mass ($m_1$). The ash content is provided by Equation (1):

$$\text{Ash content} = \frac{m_1}{m_0} \times 100 \tag{1}$$

TAPPI methods, namely, T212 om-07, T207 cm-08, and T 204-cm-07 were invested to determine the extractives in 1% NaOH, for cold and hot water, and for ethanol-toluene, respectively. Then, the $\alpha$-cellulose, Klason lignin and holocellulose were quantified using T203 cm-99, T222 om-06, and Wise et al. [39] methods, respectively. All the experiments were duplicated and the difference between the values was within and experimental error of 5%.

### 2.4. CuO Nanoparticles Biosynthesis

A total of 100 mL of 0.1 mol L$^{-1}$ copper sulphate solution (24.96 g of CuSO$_4$.5H$_2$O in 100 mL distilled water) was incorporated with 100 mL of the freshly resulting extract of the *Punica granatum* bark. The solution was agitated at 70 ± 2 °C for 3 h. The solution turned to a very dark green color. Then, after filtration, the residue was recovered and dried for 24 h at 50 °C. The resulting solid was eventually crushed (Figure 2).

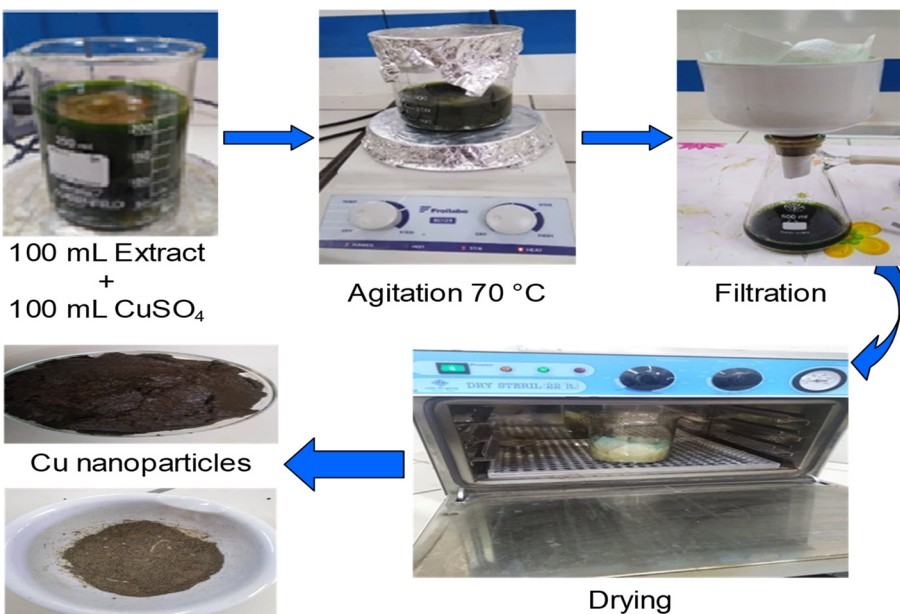

**Figure 2.** Graphical representation for CuO nanoparticles formation using extract of *Punica granatum* bark.

### 2.5. Methyl Green Adsorption on CuO Nanoparticles

The adsorption tests were carried out with a solution of methyl green. To improve the adsorbent capacity of the prepared CuO nanoparticles, we opted for the optimization of the following conditions: pH of the solution, mass of the adsorbent, initial concentration, and contact time.

The pH effect was explored through adjusting the pH in the range $2-10$ by integrating a solution of sodium hydroxide or hydrochloric acid (0.1 mol L$^{-1}$). A total of 20 mg of CuO nanoparticles was added to 10 mL of methyl green solutions (50 mg L$^{-1}$). The solutions were agitated at 120 rpm for 12 h. The specimens were then centrifuged and the methyl green concentration was specified by UV-Visible spectroscopy at a wavelength of 630 nm using a Beckman DU 800 spectrophotometer. The pH study was indicated in the form of a graphical representation of the adsorbed amount versus pH.

The impact of the adsorbent dose was traced through varying the adsorbent mass from 2 to 500 mg in 10 mL of 50 mg L$^{-1}$ methyl green solution at natural pH solution (pH = 6.5). Next, the solutions were agitated for 12 h, then centrifuged, filtered and the residual concentration was estimated. The study of the adsorbent dose corresponds to a graphical representation of the adsorbed amount versus the adsorbent dose.

To ensure good adsorption, we examined the impact of contact time on the adsorption of methyl green on CuO nanoparticles. Likewise, the previously reported steps were followed. Adsorbent (20 mg) was added to 10 mL of methyl green solution (50 mg L$^{-1}$, pH = 6.5). The solutions were stirred for a time of 20 to 420 min. The impact of contact time was obtained through a graphical representation of the quantity adsorbed versus time.

The kinetics study was carried out by following the same steps as those described previously with a solution of methyl green (50 mg L$^{-1}$). The mixtures were stirred for a time ranging from 20 to 420 min.

For adsorption isotherm, experiments were conducted by varying the initial concentration of methyl green from 5 to 120 mg g$^{-1}$ (pH = 6.5, amount of adsorbent = 20 mg, time = 60 min). The models of Langmuir [40], Freundlich [41], Temkin [42], and Dubinin–Radushkevich [43] (Table 1) were invested to model the experimental results.

**Table 1.** Theoretical models.

| Isotherm | Non-Linear Form | Linear Form |
|:---:|:---:|:---:|
| Langmuir | $q_{ads} = \dfrac{C_e q_e K_L}{1+(K_L C_e)}$ | $\dfrac{1}{q_{ads}} = \dfrac{1}{K_L q_e}\dfrac{1}{C_e} + \dfrac{1}{q_e}$ |
| Freundlich | $q_{ads} = K_F C_e^{\frac{1}{n}}$ | $\ln(q_{ads}) = \ln(K_F) + \frac{1}{n}\ln(C_e)$ |
| Temkin | $q_{ads} = \dfrac{RT}{B}\ln(A_T C_e)$ | $q_{ads} = \dfrac{RT}{B}\ln(A_T) + \dfrac{RT}{B}\ln(C_e)$ |
| Dubinin–Radushkevich | $q_{ads} = q_e e^{-\beta \varepsilon^2}$ | $\ln(q_{ads}) = \ln(q_e) - \beta \varepsilon^2$ |

$q_{ads}$ and $q_e$ correspond to the adsorbed amount and the adsorbed amount at equilibrium (mg g$^{-1}$), respectively; $C_e$ refers to the concentration of the solution at equilibrium (mg L$^{-1}$); $K_L$ and $K_F$ indicate Langmuir and Freundlich constants, respectively; (1/n) denotes Freundlich coefficient; B represents Temkin constant relating to heat of adsorption; $A_T$ signifies Temkin constant relating to adsorption potential; R expresses universal gas constant (kJ kg$^{-1}$ mol$^{-1}$ K$^{-1}$); T refers to temperature (K); β corresponds to constant related to the mean free energy of adsorption; ε stands for Polanyi potential.

For desorption experiments, the recuperated adsorbent after adsorption was recovered in 50 mL of a solution of 1.0 mol L$^{-1}$ hydrochloric acid, for 3 h. Then the sample was shaken and centrifuged and washed with distilled water. Finally, the material was dried and reused for another adsorption cycle.

### 2.6. Characterization Techniques

Scanning electron microscopy was performed out by a HITACHI S-3400N (Paris, France) device (voltage: 0.5–30 KW, resolution: 4.5 nm).

Energy dispersive X-ray spectrometry (EDS) was carried out using a Perkin Elmer DIFFRAX (Paris, France).

The thermograms were obtained by testing the samples under nitrogen flow for the temperature ranges from 30 to 950 °C for the raw material and from 30 to 600 °C for the synthesized nanoparticles (heating rate: 10 °C min$^{-1}$). The used device was the Analyzer Pyris 6 Perkin Elmer thermogravimetric analyzer (Paris, France).

The infrared analysis was conducted with a Nicolet iS 50 FT-IR Spectrometer Thermo Scientific (Paris, France). The pellets were obtained with anhydrous KBr 1% (*wt*/*wt*) of product.

## 3. Results and Discussion

### 3.1. Characterization of Punica granatum Bark

By applying standard methods, we determined the chemical composition of the bark of *Punica granatum* (Table 2).

**Table 2.** Chemical composition (content in%) of the bark of *Punica granatum* and some plant biomass collected from literature.

| | C. water | H. water | ET | 1% NaOH | Ash | Holocell. | Lignin | Cellulose |
|---|---|---|---|---|---|---|---|---|
| *Punica granatum* bark | 28 | 37.6 | 46.5 | 32.4 | 4 | 16.2 | 22 | 7.9 |
| Wood | | | | | | | | |
| Palmier Dattier [44] | n.a. | n.a. | 3 | n.a. | 6.5 | 59.5 | 27 | 33.5 |
| *Olivier* [45] | 15.5 | 17 | 10.4 | 30 | 1.4 | 65.8 | 15.6 | 41.5 |
| *Tamarisk aphylla* [46] | 12.8 | 19.0 | 4.5 | 18.5 | 3.5 | 50.0 | 30.0 | 39.0 |
| *Prunus amygdalus* [47] | 11.3 | 12.3 | 5 | 28.7 | 3.6 | 60.7 | 19.2 | 40.7 |
| Non-wood | | | | | | | | |
| Rachis de Palmier Dattier [45] | 5 | 8.1 | 6.3 | 20.8 | 5 | 74.8 | 27.2 | 45 |
| Tiges de vignes [48] | 8.2 | 13.9 | 11.3 | 37.8 | 3.9 | 65.4 | 28.1 | 35 |
| *FiguierBarbarie* [49] | 24 | 36.3 | 8.8 | 29.6 | 5.5 | 64.5 | 4.8 | 53.6 |
| *Ziziphus lotus* [50] | n.a. | n.a. | 11 | n.a. | 1.49 | n.a. | 19.6 | 30.8 |
| Annual plants | | | | | | | | |
| *Alfa* [51] | 9.1 | 11.1 | 7.9 | 19.4 | 5.1 | 69.7 | 17.71 | 47.6 |
| *Stipagrostispungens* [52] | 19.3 | 20.5 | 4.8 | 42.9 | 4.6 | 71 | 12 | 44 |
| *Astragalus armatus* [53] | 26.2 | 33 | 13 | 32.7 | 3 | 54 | 16.8 | 35 |
| *Retama raetam* [54] | 32 | 31.5 | 10 | 47 | 3.5 | 58.7 | 20.5 | 36 |
| *Nitraria retusa* [54] | 23 | 25.5 | 3 | 40 | 6.2 | 52 | 26.3 | 41 |
| *Pithuranthoschloranthus* [54] | 25 | 26.7 | 9.5 | 49 | 5 | 62 | 17.6 | 46.5 |

n.a.—not available; Holocell.—holocellulose content; 1% NaOH—content in 1% sodium hydroxide; ET—content in ethanol/toluene; H. water—hot water content; C. water—cold water content.

The bark of *Punica granatum* has an approximately high rate of extractives in cold and hot water (28%, 37.6%). These rates were higher than those of annual and perennial plants, non-wood and wood. However, extractives in a 1% NaOH solution (32.4%) were lower than those of some annual and perennial plants as well as vine stems but was higher than those of non-wood depicted in Table 2. Indeed, the bark of *Punica granatum* is rich in ester of hydrolysable ellagitannins, mainly punicalagin. Ellagitannins are high molecular weight, water-soluble polyphenols which contain several hydroxyl functions [55,56].

The extractables in the ethanol/toluene mixture displayed a rate of 46.5% which was higher than that of the materials reported in Table 2. Furthermore, the lignin content (22%) proved to be lower than that of wood plants such as date palm (27%) [44], and vine stems (28.1%) [48]. Nevertheless, it remained higher than that of olive wood (15.6%) [45], and higher than some non-wood and annual plants such as prickly pear (4.8%) [44], *Astragalus armatus* (16.8%) [53], and *Stipagrostis pungens* (12%) [52]. The holocellulose content (16.2%) and $\alpha$-cellulose (7.2%) were lower than those of non-wood, annual, and wood plants exhibited in Table 2. On the other side, the bark of *Punica granatum* presented ash content comparable to that encountered for the sources outlined in Table 2.

The infrared spectrum displayed broad stretching vibrations around 3200–3600 cm$^{-1}$ of OH groups and a band around 1360 cm$^{-1}$ reflecting alcoholic, phenolic, and acidic groups. In addition, bands of punicalagin alkyl groups appeared at 2800–2970 cm$^{-1}$. The band at 1740 cm$^{-1}$ was assigned to carbonyl groups. Moreover, the emergence of two bands

at 1620 cm$^{-1}$ and 1430 cm$^{-1}$ was attributed to the stretching of the C=C aromatic rings and the band at 1052 cm$^{-1}$ was ascribed to the vibration d stretch CO. Moreover, the bands at 1229 and 637 cm$^{-1}$ were associated respectively with the aromatic C-H bending and the C-O bond.

The atomic composition of *Punica granatum* bark powder was specified using EDX analysis. The spectrum (Figure 3) suggested the presence of carbon, oxygen, iron, sodium, potassium, and calcium.

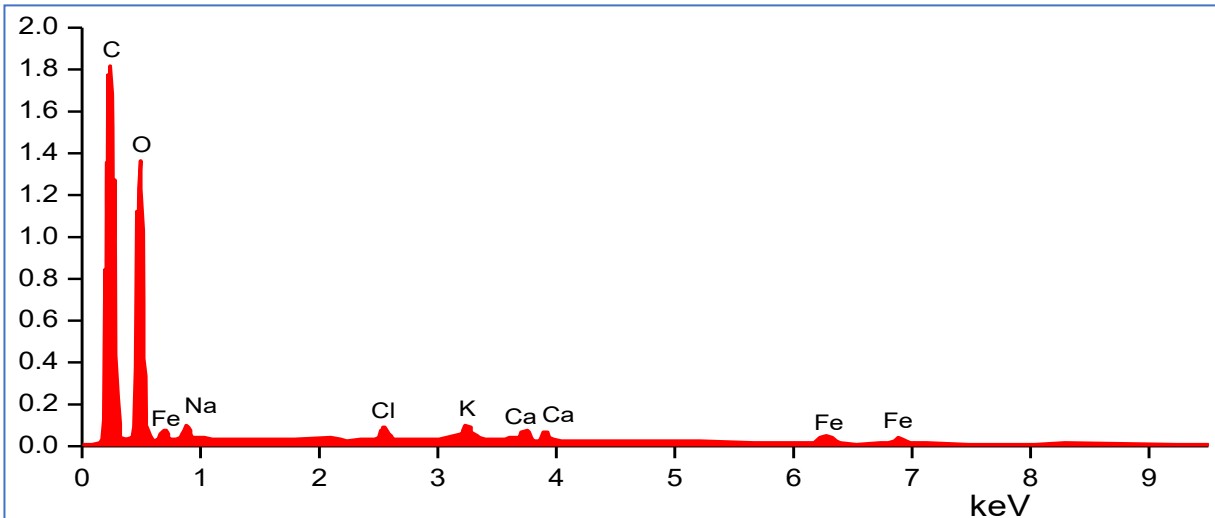

**Figure 3.** EDX analysis of *Punica granatum* bark.

The thermogram of the *Punica granatum* bark (Figure 4) revealed four stages. The first stage presented a mass loss of 5% from 30 °C to 200 °C. This loss was due to material dehydration. The second stage presented the largest mass loss (90%) occurring between 200 and 450 °C, during which the components of the lignocellulosic fibers and part of the lignin chain were degraded [57]. The third stage ranged between 450 and 600 °C. This region characterizes the thermal degradation of the remaining lignin and the complete decomposition of the carbon chains of the *Punica granatum* bark. Above 600 °C (the fourth stage) ash appeared and a constant mass was obtained.

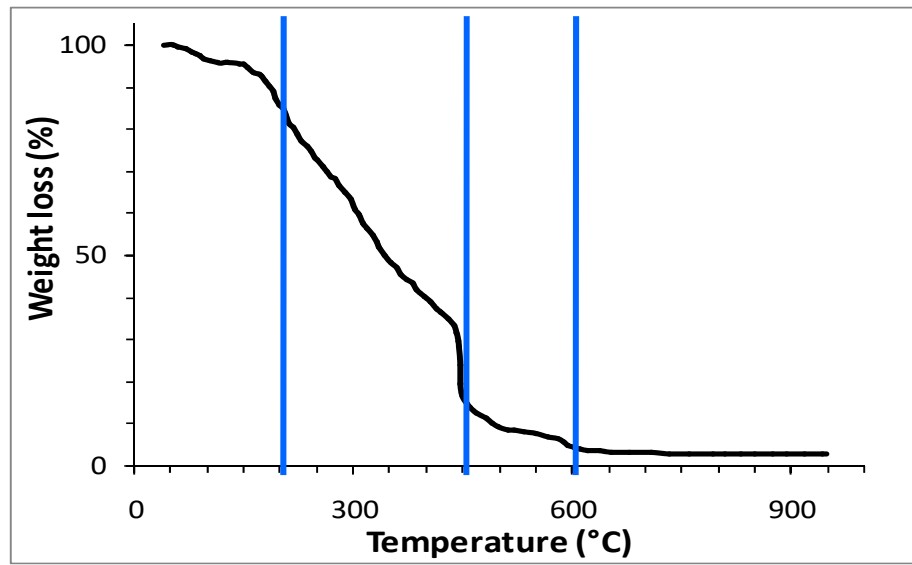

**Figure 4.** TGA curve of *Punica granatum* bark. (Blue lines indicate the three stages of mass loss).

### 3.2. Preparation and Characterization of CuO Nanoparticles

The *Punica granatum* bark is an intrinsic source of biomolecules, involving condensed tannins, ellagic acids, catechins, punicalin, and punicalagin [55,56]. These biomolecules act as ligating agents that are responsible for stabilizing and styling the copper nanoparticles. Indeed, the hydroxyl groups of the biomolecules performthe role of complexing agents that fix the copper ions of the copper sulphate solution. Although the structure of the complex formed was not determined, it canbe the product of the coordination of copper ions with the functional groups of the molecules present in the *Punica granatum* bark extract. The formed complexes underwent decomposition and led to the release of CuO nanoparticles.

The SEM images (Figure 5) of the synthesized CuO nanoparticles display a smooth surface with the arrangement of the layers on top of each other forming a multilayer structure connected with strong molecular interactions with a large variation in the distribution of the particle size and shape. This accounts for the grafting of copper ions on the organic chains of the *Punica granatum* extract. Indeed, we noticed a beam of white light reflected on the material surface referring to the response of the copper.

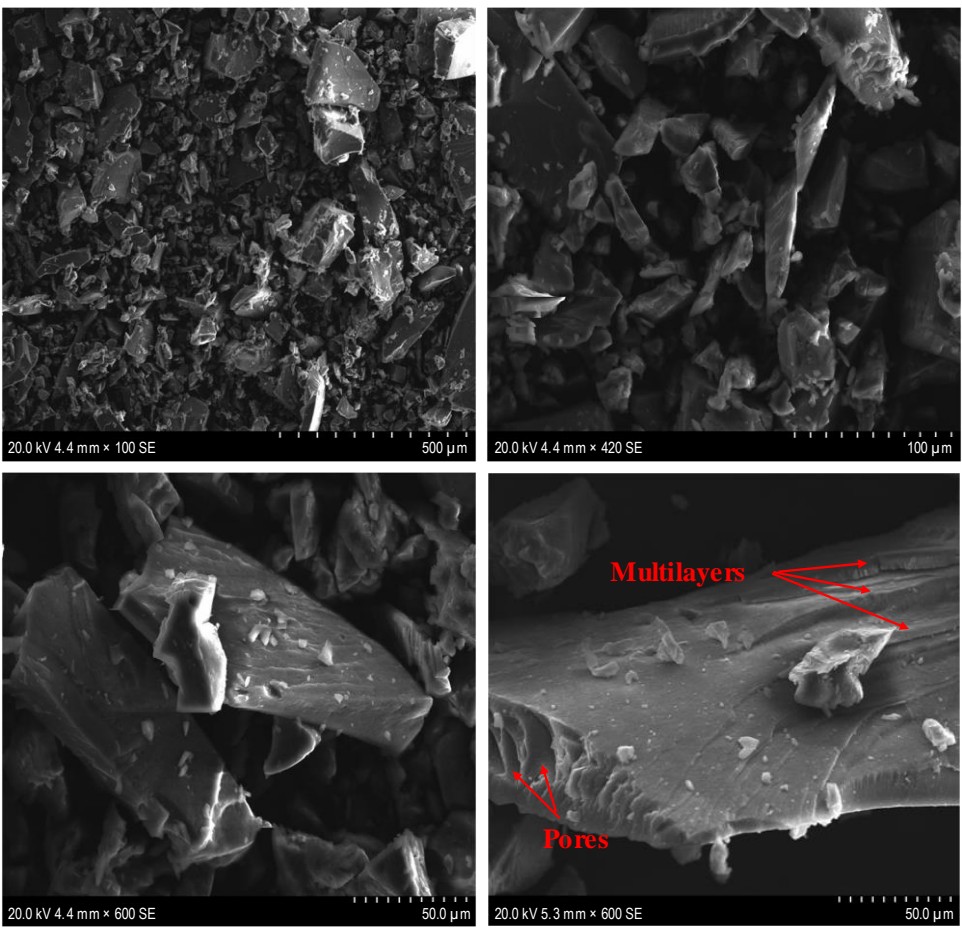

**Figure 5.** SEM microphotographs of synthesized CuO nanoparticles.

The spectrum obtained throughX-ray microanalysis (Figure 6) is indicative of peaks with strong energy for the copper found between 1 and 8 KeV, which confirms the obtaining of CuO nanoparticles. Additionally, about 54.58% of copper was quantified in the synthesized material. In addition, it is noteworthy that oxygen (19.49%) and carbon (25.93%) were noticed as constituents of the capping molecules surrounding the copper nanoparticles.

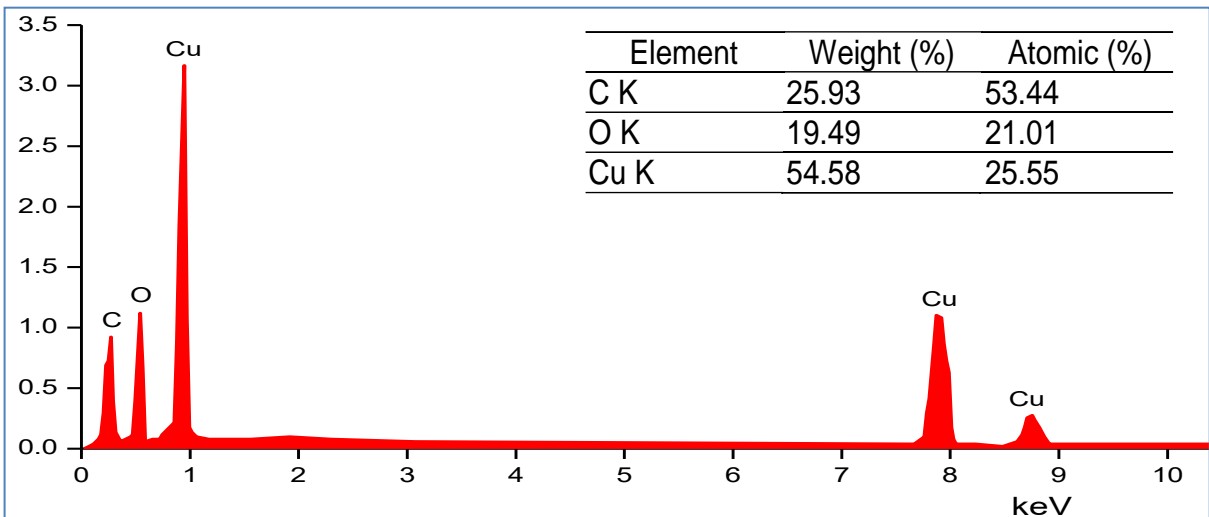

**Figure 6.** EDX analysis of CuO nanoparticles.

The thermogravimetric analysis of the synthesized CuO nanoparticles (Figure 7) revealed three main areas of mass loss: The first zone representing about 10% located between 30 and 120 °C corresponded to the release of adsorbed water. The second zone of mass loss (approximately 40%) lies between 120 and 500 °C, subdivided into two sub-regions of degradation: The first sub-region of mass loss (approximately 25%) observed between 120 and 300 °C, was attributed to the degradation and opening of the organic molecules constituting bio-capping [58], which stands for the organic support of the extract present in the prepared nanoparticles. The second sub-region (about 15%) ranging between 300 and 500 °C, was ascribed to the thermal resistance of copper which is evidenced through the slowing down of degradation processes followed by thermal decomposition of organic chain residues. The third zone located above 500 °C presented a mass stability that was recorded with a total mass loss of 51.45%. This proves that a percentage of 48.55% metallic copper was present in the prepared nanoparticles. This behavior is in good agreement with facts reported by EDX analysis.

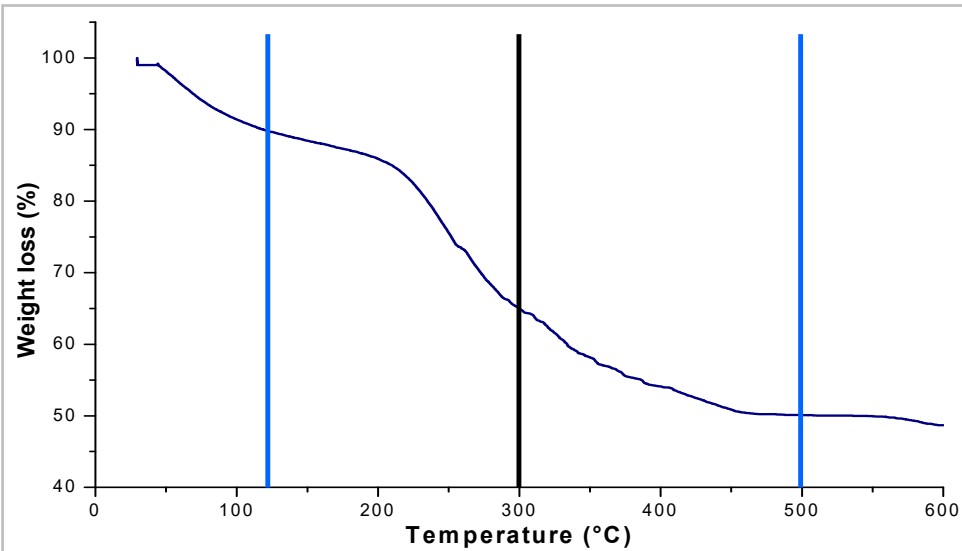

**Figure 7.** TGA curve of CuO nanoparticles. (Blue lines for main areas of mass loss and black line for sub-region of mass loss).

### 3.3. Study of the Adsorption of Methyl Green on CuO Nanoparticles

The synthesized CuO nanoparticles were used as an adsorbent for the retention of methyl green. Thus, various parameters were investigated such as methyl green concentration, contact time, adsorbent dose, and pH.

#### 3.3.1. Influence of pH, Adsorbent Dose and Contact Time on the Methyl Green Adsorption onto CuO Nanoparticles

The influence of pH is illustrated in Figure 8a. The adsorbed amount ($q_{ads}$ (mg g$^{-1}$)) increased with the pH, increasing from 2 to 6. The maximum adsorbed amount was reached for a pH ranging from 6 to 8 such as those reported using loofah fiber [59], titanium dioxide [60], CoFe$_2$O$_4$/rGO nanocomposites [61], Graphene oxide [62], activated bentonite [63], and activated carbon [64]. Indeed, when the pH value increased, many of the anionic sites were available, contributing to the increased retention of methyl green. For low pH values, H$^+$ ions were present on the surface of CuO nanoparticles, which entailed an electrostatic repulsion between the surface of the CuO nanoparticles and the molecules of methyl green, hampering the process of adsorption. Therefore, the natural pH (pH = 6.5) of the methyl green solution was used for the rest of the work.

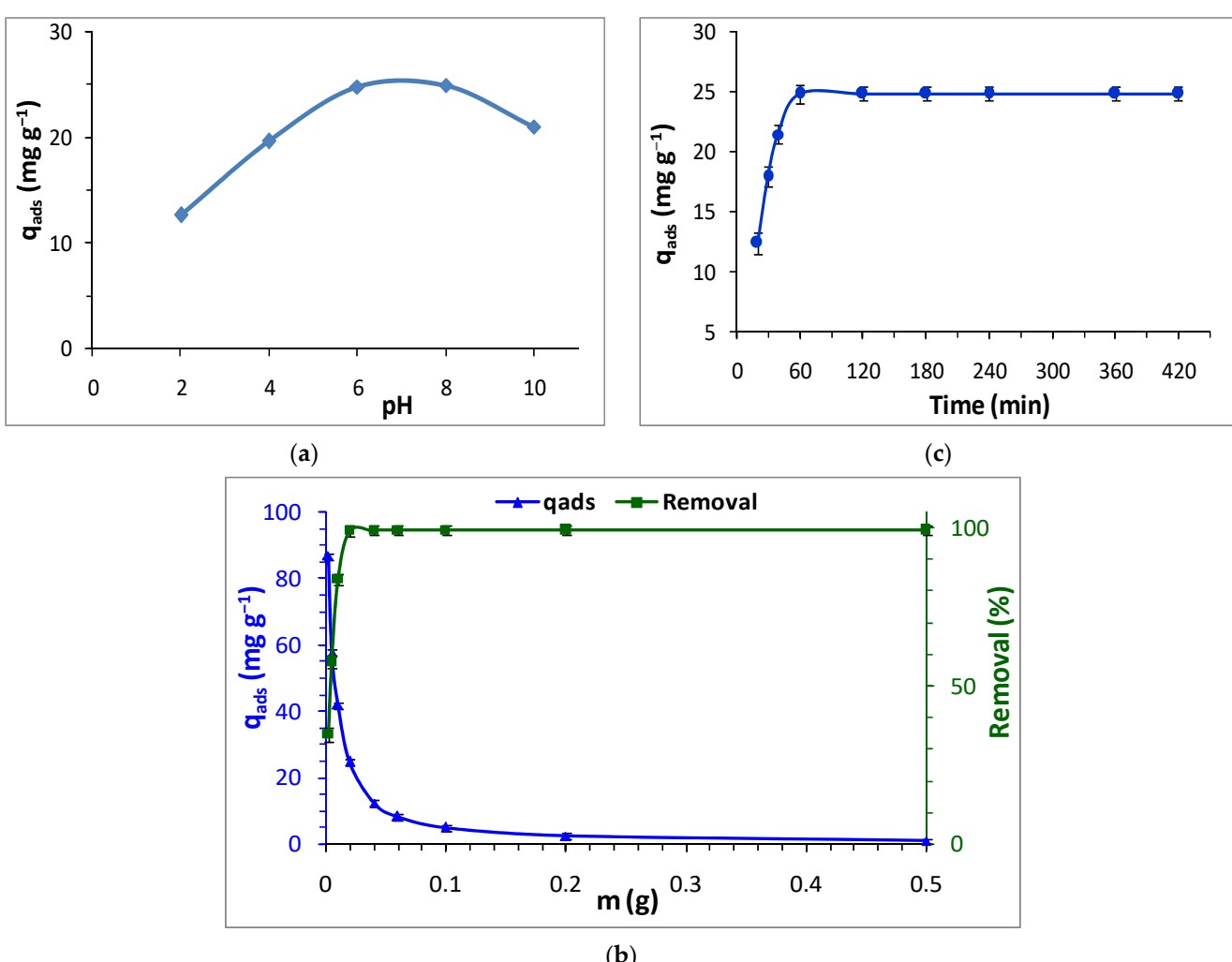

(**a**)

(**c**)

(**b**)

**Figure 8.** Influence of pH (**a**), adsorbent amount (**b**), and contact time (**c**) on the methyl green adsorption onto CuO nanoparticles. (Blue line for adsorbed quantity and green line for removal).

The adsorbed amount of methyl green decreased with an increase in adsorbent dose (Figure 8b). This is accounted for in terms of adsorption sites, which remained unsaturated

throughout the adsorption process. Thus, a mass of 20 mg of adsorbent was considered sufficient to ensure good adsorption of the methyl green on the CuO nanoparticles with complete elimination of the methyl green (elimination rate = 100%). This adsorbent mass value was selected for the current work.

To determine the time required to ensure good elimination of the methyl green, a study of the influence of the contact time on the adsorbed amount was conducted (Figure 8c). The profile of the adsorption process was characterized by two different stages. The first stage was described throughout the first 60 min of which several adsorption sites were available on the CuO nanoparticles surface. In the second stage, the elimination of methyl green remained unchanged, reflecting the saturation of the active adsorption sites. Therefore, a contact time of 60 min was considered sufficient to ensure adsorption equilibrium.

### 3.3.2. Kinetic Study

Both the pseudo-first order Equation (2) [65–67] and pseudo-second order Equation (3) [66–68] models were invested to characterize the kinetics of adsorption of methyl green on CuO nanoparticles.

$$\ln(q_e - q_t) = \ln q_e - k_1 t \tag{2}$$

$$\frac{t}{q_t} = \frac{1}{k_2 q_e^2} + \frac{t}{q_t} \tag{3}$$

where $q_e$ and $q_t$ (mg g$^{-1}$) refer, respectively, to the adsorbed quantities at equilibrium and at time "t"; $k_1$ and $k_2$ represent the constants for pseudo-first and pseudo-second order model, respectively.

The value of the adsorbed amount of methyl green calculated ($q_{cal}$ = 25.6 mg g$^{-1}$) using the pseudo-second order model was comparable to the experimental value ($q_{exp}$ = 24.9 mg g$^{-1}$) (Table 3). Additionally, the regression coefficient value in the case of the pseudo-second order model ($R^2$ = 0.999) was high in comparison with that recorded for the pseudo-first order model. This suggests that the adsorption of methyl green on CuO nanoparticles adheres to the pseudo-second order kinetics. Likewise, according to $\Delta q$, RMSE and $\chi^2$, the pseudo-second order model displays lower errors and is in good consistency with the experimental data. Similar phenomena have been reported for the adsorption of methyl green onto various adsorbent such as loofah fiber [59], titanium dioxide [60], zeolite H-ZSM-5 [69], carbon nanotubes [70], and graphene sheets [71].

**Table 3.** Kinetics parameters of methyl green adsorption on CuO nanoparticles.

|  | $q_{exp}$ (mg g$^{-1}$) | **24.9** |
|---|---|---|
| Pseudo-first order | $q_{cal}$ (mg g$^{-1}$) | 7.0 |
|  | $k_1$ (min$^{-1}$) | 0.0287 |
|  | $R^2$ | 0.7911 |
|  | $\chi^2$ | 232.297 |
|  | $\Delta q$ | 0.5199 |
|  | RMSE | 15.2870 |
| Pseudo-second order | $q_{cal}$ (mg g$^{-1}$) | 25.6 |
|  | $k_2$ (g mg$^{-1}$ min$^{-1}$) | 0.0047 |
|  | $R^2$ | 0.9990 |
|  | $\chi^2$ | 8.238 |
|  | $\Delta q$ | 0.1125 |
|  | RMSE | 4.8492 |

### 3.3.3. Adsorption Isotherms

The Temkin, Langmuir, Freundlich, and Dubinin–Radushkevich isotherms were used to characterize the adsorption of methyl green on the prepared nanoparticles (Table 4).The

goodness of fit of the theoretical models at the experimental isotherm was confirmed by comparing the values of the regression coefficients $R^2$.

**Table 4.** Isothermal parameters.

| | | |
|---|---|---|
| Langmuir | $q_{max}$ (mg g$^{-1}$) | 28.7 |
| | $K_L$ (L mg$^{-1}$) | 6.96 |
| | $R_L$ | 0.0011–0.0279 |
| | $R^2$ | 0.9892 |
| | $\chi^2$ | 1.9458 |
| | $\Delta q$ | 0.0144 |
| Freundlich | $1/n$ | 0.2294 |
| | $K_F$ | 16.683 |
| | $R^2$ | 0.7084 |
| | $\chi^2$ | 12.5882 |
| | $\Delta q$ | 0.2419 |
| Temkin | $R^2$ | 0.8699 |
| | $B$ (J mol$^{-1}$) | 3.3198 |
| | $A_T$ (L g$^{-1}$) | 424.766 |
| | $\chi^2$ | 5.2396 |
| | $\Delta q$ | 0.1960 |
| Dubinin–Radushkevich | $R^2$ | 0.9649 |
| | $B$ | 0.0215 |
| | $q_{max}$ (mg g$^{-1}$) | 27.709 |
| | $\chi^2$ | 6.7088 |
| | $\Delta q$ | 0.0217 |

It seems that the Langmuir fit ($R^2 = 0.9892$) fits the methyl green adsorption data better than the other three models. It turns out that the Langmuir model is the most adequate to characterize the adsorption of methyl green on CuO nanoparticles. This result was checked by the low value of the chi-square test $\chi^2$, and the standard deviation $\Delta q$, reflecting monolayer adsorption. Moreover, the values of dimensionless equilibrium constant $R_L$, (Equation (4)) were lower than 1, reflecting the favorable adsorption of methyl green on the CuO nanoparticles. This fact was confirmed by the value of the Freundlich coefficient ($1/n = 0.2294$), which was lower than 1 [67,72–74].

$$R_L = \frac{1}{1 + K_L C_0} \tag{4}$$

The maximum adsorbed amount of methyl green on CuO nanoparticles calculated from the Langmuir model, was 28.7 mg g$^{-1}$.

A comparison of the adsorption capacity of methyl green onto CuO nanoparticles with other adsorbents reported by other researchers was shown in Table 5. It can be seen that CuO nanoparticles showed comparable adsorption capacity with respect to zeolite H-ZSM-5 [69] and grapheme oxide [62]. The adsorption capacity of CuO nanoparticles was higher than those of loofah fiber (18.2 mg g$^{-1}$) [59] and bamboo [75], and was lower than that of titanium dioxide (384.6 mg g$^{-1}$) [60], activated carbon from *Brachychiton Populneus* fruit shell [64], activated bentonite [63], and graphene sheets [71]. Therefore, the CuO nanoparticles were suitable and promising for the removal of methyl green from aqueous solutions since they have a relatively higher adsorption capacity.

**Table 5.** Reported maximum adsorption capacities in the literature for methyl green adsorption.

| Adsorbent | Adsorption Conditions | $q_{max}$ (mg g$^{-1}$) |
|---|---|---|
| Loofah fiber [59] | pH = 6.5; dose = 0.4 g/L; T = 25 °C | 18.2 |
| Bamboo [75] | pH of solution; dose =1 g/L; T = 25 °C | 20.4 |
| Graphene oxide [62] | pH = 7; dose = 0.1 g/L; T = 25 °C | 28.5 |
| Zeolite H-ZSM-5 [69] | pH = 4; dose = 1 g/L; T = 25 °C | 31.3 |
| Graphene sheets-CoFe$_2$O$_4$ nanparticles [71] | pH of solution; dose = 1 g/L; T = 25 °C | 47.2 |
| Activated carbon from *Brachychiton Populneus* fruit shell [64] | pH of solution; dose = 1 g/L; T = 20 °C | 67.9 |
| CoFe$_2$O$_4$/rGO nanocomposites [61] | pH of solution; dose = 0.25 g/L | 88.3 |
| (CNTs-NiFe$_2$O$_4$) [70] | pH of solution; dose = 1 g/L; T = 25 °C | 88.5 |
| CNTs [70] | pH of solution; dose = 1 g/L; T = 25 °C | 146 |
| Graphene sheets [71] | pH of solution; dose = 1 g/L; T = 25 °C | 203.5 |
| Activated bentonite [63] | pH of solution; dose = 0.1 g/L; T = 25 °C | 335.3 |
| Titanium dioxide [60] | pH = 6.6; dose = 1 g/L; T = 25 °C | 384.6 |
| CuO nanoparticles (this work) | pH of solution; dose = 2 g/L; T = 25 °C | 28.7 |

### 3.3.4. Desorption and Adsorbent Reuse

To assess the feasibility and applicability of adsorbent reuse and therefore to increase the use effectiveness of the prepared nanoparticles, desorption of the adsorbed methyl green was performed. Departing from the study of the pH effect, it was demonstrated that adsorption of methyl green was defective in acidic medium. Thus, desorption was conducted by washing the recuperated adsorbent after adsorption with a hydrochloric acid and afterwards with distilled water. Next, the adsorbent was dried and reused for the adsorption of methyl green. The adsorption/desorption procedure was repeated five times and results were summarized in Table 6.

**Table 6.** Adsorption-desorption of methyl green.

| Cycle | $q_{ads}$ (mg g$^{-1}$) | Adsorption (%) | Desorption (%) |
|---|---|---|---|
| 1 | 24.9 | 99.8 | 99.2 |
| 2 | 24.3 | 97.3 | 98.6 |
| 3 | 23.1 | 92.4 | 94.1 |
| 4 | 20.8 | 83.3 | 93.2 |
| 5 | 17.1 | 68.4 | 90.4 |

The maximum adsorption capacity of CuO nanoparticles decreased from 24.9 mg g$^{-1}$ to 17.1 mg g$^{-1}$ after five cycles. The regeneration results revealed that the adsorption efficiency of CuO nanoparticles dropped of only about 8% after three cycles. The adsorption/desorption tests corroborated appropriate reuse characteristics of the adsorbent after three cycles with a better performance for methyl green sorption.

### 4. Conclusions

The green synthesis of metallic nanoparticles has accumulated interest over the past decade due to their distinctive properties that make them applicable in various fields. Metal nanoparticles synthesized using plants have been shown to be non-toxic and environmentally friendly. In this study, a very cheap and simple conventional method was used to obtain the CuO nanoparticles using the extract from the bark of *Punica granatum*. The nanoparticles' purity was confirmed through EDX, yielding the presence of elemental copper oxide containing 25.93, 19.49, and 54.58%, of carbon, oxygen, and copper, respectively. The synthesized CuO nanoparticles demonstrated a great performance in the adsorption of methyl green with an adsorption capacity of 28.7 mg g$^{-1}$ under optimal conditions (60 min, contact time; 2 g L$^{-1}$, adsorbent dose; pH, 6.5; 25 ± 2 °C). The adsorption of methyl green on the CuO nanoparticles follows the pseudo-second order model and the adsorption isotherm is in good conformity with the Langmuir model. CuO nanoparticles can be reused with good recycle properties for three cycles. Regarding the good sorption performance of

the prepared CuO nanoparticles, the use of environmentally friendly *Punica granatum* bark can constitute an ecological, simple, and economical method for the elaboration of materials in order not only to purify contaminated waters but also to enact additional environmental applications. The nanoparticles obtained through plant extracts have the potential to be widely used in the medical field as dressings or therapeutic drugs. Thus, green chemistry provides a new innovative technique that does not use hazardous substances for the design and development of variable activity materials.

**Author Contributions:** Literature review, data collection, field work, M.b.M.; methodology and writing—original draft preparation, M.b.M. and Y.M.; designed and revised the manuscript, L.M. and F.A.; writing—review and editing, A.K.D.A.; visualization and supervision, Y.M. All authors have read and agreed to the published version of the manuscript.

**Funding:** This research was funded by scientific research deanship at University of Ha'il—Saudi Arabia through project number RG-191251. The APC was funded by scientific research deanship at University of Ha'il—Saudi Arabia through project number RG-191251.

**Institutional Review Board Statement:** Not applicable.

**Informed Consent Statement:** Not applicable.

**Data Availability Statement:** Not applicable.

**Acknowledgments:** This research has been funded by scientific research deanship at University of Ha'il—Saudi Arabia through project number RG-191251.

**Conflicts of Interest:** The authors declare no conflict of interest.

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
