# Peer review of "Ecological Synthesis of CuO Nanoparticles Using Punica granatum L. Peel Extract for the Retention of Methyl Green"

_water, doi:10.3390/w14091509_

Round 1

Reviewer 1 Report

In this work, CuO−nanoparticles is prepared by the  bark extract of Punica granatum L. and the adsorption performance of  a green methyl green are evaluated. It is an economical, ecological and clean nanoparticle preparation method. But the mechanism study is normal and simple, the innovation of this work is weak. Meanwhile, the introduction, conclusion, and discussion part need careful revision. Thus, major revision is suggested for this manuscript.

  1. Introduction: This part is so simple, which had not cover the key research point in this work. More detailed research status about the CuO preparation or methyl green retention technology was suggested.
  2. Line 71: filtered by what?
  3. Part 2.3: need revision, the methods were not clear. And some paragraphs can be merged.
  4. Part 3.1: what is the purpose for the comparison of the composition of PG bark and other plant? Or, did the composition of PG bark had advantage for CuO synthesis?
  5. How dose the  composition of PG bark extractive on the CuO?
  6. Line 222-223, if the decomposition started from 30 °C, how can it be effectvely used in water treatment? Or, the decomposition may bring new pollutants for water?
  7. Is it possible to call it CuO nanoparticles as the content was about 50%? Is CuO@biochar more suitable?
  8. For the adsorption study, normal analysis was applied for adsorption mechanism discussion, more measurement  methods are needed to explain the adsorption performance of CuO, i.e., XRD, XPS, FTIR, etc.
  9. A comparison study of the performance of this work with others should be added.
  10. The conclusion part should be carefully revised.
  11. What is the innovation of this work? Please make it clear.

Author Response

Dear colleague,

First of all, I would like to thank you for the time you are spending to review our paper. We tried to take into account all the remarks. The corresponding corrections in the revised manuscript are highlighted with a yellow background, and the replies to your queries are listed below.

In this work, CuO−nanoparticles is prepared by the bark extract of Punica granatum L. and the adsorption performance of a green methyl green are evaluated. It is an economical, ecological and clean nanoparticle preparation method. But the mechanism study is normal and simple, the innovation of this work is weak. Meanwhile, the introduction, conclusion, and discussion part need careful revision. Thus, major revision is suggested for this manuscript.

  1. Introduction: This part is so simple, which had not cover the key research point in this work. More detailed research status about the CuO preparation or methyl green retention technology was suggested.

Reply:

Thank you for your comments. We have revised the introduction and have added more details about CuO preparation and methyl green retention technology. We have included the proposed articles in the revised manuscript and summarized its main finding relevant to our investigation.

  1. Line 71: filtered by what?

Reply:

Thank you. The mixture was filtered using Whatman filter paper no.01.

  1. Part 2.3: need revision, the methods were not clear. And some paragraphs can be merged.

Reply:

Thank you for your comment. We have revised this part and we have merged paragraphs. The used methods are standard methods.

  1. Part 3.1: what is the purpose for the comparison of the composition of PG bark and other plant? Or, did the composition of PG bark had advantage for CuO synthesis?

Reply:

Thank you for your comments. In our study, we have opted to present the chemical composition of PG bark compared with some other plants. As mentioned in section 3.2. PG bark extract was characterized by its richness of biomolecules such as condensed tannins, ellagic acids, catechins, punicalin, and punicalagin. It appears that these biomolecules enhanced the formation of CuO particles. Thus, the aqueous extract of PG bark provides both a reducing and stabilizing agent for the biosynthesis of CuO nanoparticles

  1. How dose the composition of PG bark extractive on the CuO?

Reply:

Thank you. The extract of PG bark used was freshly prepared as mentioned in experimental section. We have prepared the extract using 200g of PG bark and 800 Ml of water. Then, the synthesis was conducted using equal volumes of freshly prepared extract and a 0.1 mol L-1 CuSO4 solution.

  1. Line 222-223, if the decomposition started from 30 °C, how can it be effectvely used in water treatment? Or, the decomposition may bring new pollutants for water?

Reply:

Thank you for this remark. As mentioned in section 3.2 (page 8), the decomposition starting from 30°C to 120°C corresponds to the release of adsorbed water, and not for a decomposition of CuO nanoparticles.

  1. Is it possible to call it CuO nanoparticles as the content was about 50%? Is CuO@biochar more suitable?

Reply:

Thank you. In the manuscript, nanoparticles prepared are already called CuO-nanoparticles and it is possible to call copper nanoparticles. However, CuO@biochar is not used in the literature and there is no pyrolysis step to obtain biochar.

  1. For the adsorption study, normal analysis was applied for adsorption mechanism discussion, more measurement methods are needed to explain the adsorption performance of CuO, i.e., XRD, XPS, FTIR, etc.

Reply:

Thank you for your valuable comment. Yes it is important to present other analysis of adsorbent after adsorption. At this moment, we limited our study based on studying the evolution of the concentration of the adsorbate to establish the adsorption performance of CuO nanoparticles. However, we thank you for the idea and XRD, XPS analysis will be the subject of our future works.

  1. A comparison study of the performance of this work with others should be added.

Reply:

Thank you for this suggestion. We have added a comparison with others results from literature in the revised manuscript.

  1. The conclusion part should be carefully revised.

Reply:

Thank you. We have restructured and rewritten the conclusion in the revised manuscript.

  1. What is the innovation of this work? Please make it clear.

Reply:

Thank you for this remark. This work aims at the use of an ecological method for the elaboration of copper nanoparticles from the extract of the bark of Punica granatum L. The obtained CuO−nanoparticles were used successfully for removal of methyl green from aqueous solution. Thus, the innovation of this work is illustrated through the use of an ecological and reproducible synthesis of nanoparticles using plant extracts.

Reviewer 2 Report

  1. The number of significant digits should be significantly reduced, e.g. 12.5882 can be written as 12.6, 28.664 should be written as 28.7 and 424.766 is 424.8 - this should be corrected throughout the manuscript.
  2. Why did the experimental studies on the effect of pH use a 10 mg adsorbent mass and the other studies use a 20 mg adsorbent mass. All studies should be performed under the same conditions. This part of the study should be repeated for the 20 mg sample.
  3. Desorption was not described in the experimental part of the paper - this needs to be completed. How much dye was adsorbed in the first sorption cycle and how much was desorbed in the first desorption cycle. These data must be given for each of the five cycles.
  4. 28,664 mg g−1  or 28.664 mg g−1  ???? Please check carefully - this is a very big discrepancy.
  5. The main shortcoming of the paper is the lack of comparison between the experimental data obtained and those published in the literature. The dye used by the research team is very common in model studies. The results of the kinetic experiments and the values of the obtained sorption capacity should be compared with the results published by other researchers (concerning the removal of methyl green on other or similar adsorbents). A similar observation applies to pH.

Author Response

Dear colleague,

First of all, I would like to thank you for the time you are spending to review our paper. We tried to take into account all the remarks. The corresponding corrections in the revised manuscript are highlighted with a yellow background, and the replies to your queries are listed below.

  1. The number of significant digits should be significantly reduced, e.g. 12.5882 can be written as 12.6, 28.664 should be written as 28.7 and 424.766 is 424.8 - this should be corrected throughout the manuscript.

Reply:

Thank you for your suggestion. The suggestion was considered in the revised manuscript.

  1. Why did the experimental studies on the effect of pH use a 10 mg adsorbent mass and the other studies use a 20 mg adsorbent mass. All studies should be performed under the same conditions. This part of the study should be repeated for the 20 mg sample.

Reply:

Thank you for pointing this out. Sorry for this mistake, the effect of pH and other parameters studied were conducted using 20 mg of adsorbent. This was corrected in the revised manuscript.

  1. Desorption was not described in the experimental part of the paper - this needs to be completed. How much dye was adsorbed in the first sorption cycle and how much was desorbed in the first desorption cycle. These data must be given for each of the five cycles.

Reply:

Thank you for your comments. We have added more details about desorption experiments in the experimental part. We have added more data about adsorption and desorption experiments.

  1. 28,664 mg g−1  or 28.664 mg g−1  ???? Please check carefully - this is a very big discrepancy.

Reply:

Thank you for your comment. The values have been checked and corrected in the revised manuscript.

  1. The main shortcoming of the paper is the lack of comparison between the experimental data obtained and those published in the literature. The dye used by the research team is very common in model studies. The results of the kinetic experiments and the values of the obtained sorption capacity should be compared with the results published by other researchers (concerning the removal of methyl green on other or similar adsorbents). A similar observation applies to pH.

Reply:

Thank you for this suggestion. We have added a comparison with others results from literature in the revised manuscript.

Round 2

Reviewer 1 Report

  1. And some paragraphs can be merged as all figures were simple.
  2. As CuO just account for 50% of the complex material, it was not suitble to call it CuO nanoparticles
  3. A Table to compare the performance of this work with others should be added.
  4. The innovation of this work is weak, and which was not enhanced during the revision.

Author Response

Dear colleague,

Thank you for taking the time to review our paper and for your valuable comments and suggestions, they were very insightful.

We tried to take into account all comments. Below, you will find a point by point description of how each comment is addressed in the manuscript. The corresponding corrections in the revised manuscript are highlighted with a blue background.

  1. And some paragraphs can be merged as all figures were simple..

Reply:

Thank you for your comments. We have merged some paragraphs (3.3.1.).

  1. As CuO just account for 50% of the complex material, it was not suitble to call it CuO nanoparticles.

Reply:

Thank you. The synthesized material was composed of 54.58% of copper, 19.49 % of oxygen and 25.93 % of carbon. The composition of the obtained material was comparable with previously mentioned in the literature and this material was called CuO-nanoparticles or copper nanoparticles. (Examples: Open Nano 3 (2018) 18–27; JOM 72 (2020) 1264–1272; Current Biotechnology 9 (2020) 304–311; Malaysian Journal of Fundamental and Applied Sciences 15 (2019) 218-224; Current Research in Green and Sustainable Chemistry 5 (2022) 100249; Environmental Nanotechnology, Monitoring & Management 14 (2020) 100346; Arabian Journal of Chemistry (2020) 13, 4263–4274; …). Please if you have any proposition.

  1. A Table to compare the performance of this work with others should be added.

Reply:

Thank you for your comment. We have added a table summarising some others results from literature and compared to obtained results in this work.

  1. The innovation of this work is weak, and which was not enhanced during the revision.

Reply:

This work aims at the use of an ecological method for the elaboration of CuO−nanoparticles from the extract of the bark of Punica granatum L. Thus, the innovation of this work is illustrated through the use of an ecological and reproducible synthesis of nanoparticles using plant extracts. The nanoparticles obtained through plant extracts have the potential to be widely used in the medical field as dressings or therapeutic drugs. Thus, green chemistry provides a new innovative technique that does not use hazardous substances for the design and development of variable activity materials.

Reviewer 2 Report

  1. Unfortunately, the authors have not taken all the comments into account in a proper and satisfactory manner. Point 5 of the review has been ignored. A Table of equilibrium data (sorption capacity, weighting, adsorbate volume, temperature, pH) should be prepared for at least 8-10 adsorbents.
    Providing two examples unfortunately does not meet my expectations.
    Furthermore, the comparison of kinetic data is not included - no comparison of kinetic data with those available in the literature - this answer was completely ignored. Comparison of pH is another aspect that was ignored.
  2. In the tables concerning kinetic and equilibrium parameters, the units must be included; this will make it much easier for the reader to perceive the published data without having to search the whole paper for units.
  3. The data in Figure 9 should be transferred to Table 5 - just add two columns.

Author Response

Dear colleague,

Thank you for taking the time to review our paper and for your valuable comments and suggestions, they were very insightful.

We tried to take into account all comments. Below, you will find a point by point description of how each comment is addressed in the manuscript. The corresponding corrections in the revised manuscript are highlighted with a blue background.

  1. Unfortunately, the authors have not taken all the comments into account in a proper and satisfactory manner. Point 5 of the review has been ignored. A Table of equilibrium data (sorption capacity, weighting, adsorbate volume, temperature, pH) should be prepared for at least 8-10 adsorbents.
    Providing two examples unfortunately does not meet my expectations.
    Furthermore, the comparison of kinetic data is not included - no comparison of kinetic data with those available in the literature - this answer was completely ignored. Comparison of pH is another aspect that was ignored.

Reply:

Thank you for your comments. We have added a comparison with others results from literature in the revised manuscript.

  1. In the tables concerning kinetic and equilibrium parameters, the units must be included; this will make it much easier for the reader to perceive the published data without having to search the whole paper for units.

Reply:

Thank you for your comments. We have included the units in the tables.

  1. The data in Figure 9 should be transferred to Table 5 - just add two columns.

Reply:

Thank you for your suggestion. We have deleted figure 9 and data were transferred to table 5.

Round 3

Reviewer 2 Report

The article can be published in the current form.

This manuscript is a resubmission of an earlier submission. The following is a list of the peer review reports and author responses from that submission.